Assessment of activities of daily living in patients post COVID-19: a systematic review

http://orcid.org/0000-0002-0248-3312 Pizarro-Pennarolli Catalina 1
Sánchez-Rojas Carlos 1
Torres-Castro Rodrigo 1 2
Vera-Uribe Roberto 1 2
http://orcid.org/0000-0003-1637-4309 Sanchez-Ramirez Diana C. 3
http://orcid.org/0000-0003-0405-3831 Vasconcello-Castillo Luis 1 2
Solís-Navarro Lilian 1
Rivera-Lillo Gonzalo 1 4 5 gbrivera@uchile.cl
1 Department of Physical Therapy, Faculty of Medicine, Universidad de Chile , Santiago , Chile
2 International Physiotherapy Research Network (PhysioEvidence) , Barcelona , España
3 Department of Respiratory Therapy, College of Rehabilitation Sciences, University of Manitoba , Winnipeg, MB , Canada
4 Research and Development Unit, Clínica Los Coihues , Santiago , Chile
5 Department of Neuroscience, Faculty of Medicine, Universidad de Chile , Santiago , Chile
Morris Linda
Electronic publication date: 2021 Apr 6
Publication date: 2021
Volume: 9
Electronic Location ID: e11026
Received 2020 Nov 23; Accepted 2021 Feb 8
Copyright: © 2021 Pizarro-Pennarolli et al.
Copyright year: 2021
Copyright holder: Pizarro-Pennarolli et al.
License: This is an open access article distributed under the terms of the Creative Commons Attribution License, which permits unrestricted use, distribution, reproduction and adaptation in any medium and for any purpose provided that it is properly attributed. For attribution, the original author(s), title, publication source (PeerJ) and either DOI or URL of the article must be cited.
License URL: https://creativecommons.org/licenses/by/4.0/

Keywords: COVID-19, Rehabilitation, Activities of daily living, Functional assessment, Systematic review

Funding: National Agency for Research and Development (ANID) 2018-72190117 Rodrigo Torres-Castro is funded by a grant from the National Agency for Research and Development (ANID)/Scholarship Program/DOCTORADO BECAS CHILE/2018-72190117. The funders had no role in study design, data collection and analysis, decision to publish, or preparation of the manuscript.

==============================
Background

Coronavirus disease has provoked much discussion since its first appearance. Despite it being widely studied all over the world, little is known about the impact of the disease on functional ability related to performing activities of daily living (ADL) in patients post COVID-19 infection.

Objectives

To understand the impact of COVID-19 on ADL performance of adult patients and to describe the common scales used to assess performance of ADL on patients post-COVID-19.

Methods

A systematic review was conducted. We included studies that applied a physical capacity test in COVID-19 patients, post-infection. Two independent reviewers analyzed the studies, extracted the data, and assessed the quality of the evidence.

Results

A total of 1,228 studies were included, after removing duplicates, 1,005 abstracts were screened and of those 983 were excluded. A final number of nine studies which met the eligibility criteria were included. The findings revealed worsening of physical function and ADL performance in all patients post COVID-19 infection.

Conclusion

All included studies found a reduction of ADL beyond the test or scale used, revealing a vital worsening of functional ability in ADL performance and consequently loss of independence in COVID-19 patients after the acute phase of infection. Functional ability status previous to COVID-19 is crucial for predicting the severity of the disease and mortality. Barthel Index and ADL score were the most used assessment tools across subjects with different intrinsic capacity and context levels.

Introduction

In December 2019 a SARS-COV-2 virus emerged in Wuhan, China, which has rapidly spread around the world (Phelan, Katz & Gostin, 2020). In March 2020, the World Health Organization (WHO) declared a global pandemic due to the disease caused by this novel coronavirus (COVID-19) (Adhikari et al., 2020).

The magnitude of the sequelae produced by the disease at different functioning levels are still under research. The consequences, such as fatigue and breathlessness, have been reported to persist after hospital discharge and the impact at different functioning levels over time remains unclear (Carfì, Bernabei & Landi, 2020; Halpin et al., 2020). These findings are in line with the presence of symptoms such as myalgia that has been reported in about 15% of patients 4 months post-COVID-19 infection (Garrigues et al., 2020).

The immediate and short-term assessment of physical capacity in patients with COVID-19 has been a challenge, with a variable number of subjects being able to perform the tasks included on the most commonly used physical capacity assessment tests (Belli et al., 2020; Curci et al., 2020; Sakai et al., 2020; Liu et al., 2020; Bousquet et al., 2020; Zerah et al., 2020). In this way, it seems necessary and reasonable to complement the clinical assessment of physical capacity with a further analysis of functional ability related to perform activities of daily living (ADL).

Impairments in body functions and structure, such as dyspnea, weakness, myalgia and other pain sensations limit the ability to perform both basic ADL (BADL) and instrumental ADL (IADL). BADL are related to personal care and mobility (such as dressing, eating, ambulating, toileting, hygiene) whereas IADL is associated with the person’s ability to interact with his/her environment. Limitations in the ability to perform ADL increase the susceptibility to care dependency, reducing the quality of life of both the affected person and the caregiver (Dunsky, 2019). Dependence tends to be accentuated in the elderly population and through all age ranges after an extended ICU stay (Herridge et al., 2016); a reduced level of ADL independence has been related to worse prognosis after COVID-19 (Bousquet et al., 2020; Zerah et al., 2020).

The WHO consider two main aspects when defining functional ability. First, the intrinsic capacity related to all the physical and mental capacities of an individual. Second, the environment related to all the factors in the extrinsic world that form the context of an individual’s life (World Health Organization, 2015). In this way, functional ability during ADL is determined by the impairment affecting the intrinsic capacity as consequence of COVID-19, and the environment is determined by the context of each individual’s life, including variables such as the time course of the disease or the previous functional status. Measuring ADL is relevant since it provides an essential framework of the individual’s current functional state, and thus the requirements for rehabilitation and other resources (Rivera-Lillo et al., 2020).

Scales commonly used to assess functional status regarding ADL in inpatients and survivors of critical illness are: Barthel Index (BI), Functional Independence Measure (FIM), Modified Rankin Scale (mRS), EQ-5D, among others (Elliott et al., 2011). The mRS has been widely used as the primary outcome measure for acute stroke (Broderick, Adeoye & Elm, 2017), whereas FIM was identified as a less biased assessment with greater reliability and validity compared to other instruments (BI or Katz) (Cohen & Marino, 2000). Most of the scales allow quantification of the magnitude of functional independence, based on the assistance requirements across different items related to activities or participation during ADL.

The combination of variables leading to different scenarios related to intrinsic capacity and environment places different limitations on ADL. It is therefore necessary to detect the best way to know and understand functional ability during ADL before and after COVID-19. To fill this knowledge gap, in this work we analyze the most common test reported to date to evaluate the relationship between ADL functional ability and COVID-19. We also describe the factors related to context and intrinsic capacity to provide an overview of the necessity to assess the impact of COVID-19 in ADL limitations. Appropriate assessment scales will help to objectively evaluate the effect of COVID-19 on the functional ability of patients over time, leading to optimal care and rehabilitation strategies for affected subjects. At the same time, ADL functional ability status previous to COVID-19 provides a fine-grained understanding of the importance of individual context in the ADL functional status post-COVID-19. The aim of this report is to understand the impact of COVID-19 on ADL performance of adult patients and to describe the common scales used to assess ADL functional ability status on patients before and after COVID-19.

Materials and Methods

We conducted a systematic review of the literature according to the Preferred Reporting Items for Systematic Reviews and Meta-Analyses (PRISMA) guidelines (Moher et al., 2009). The review was registered in the International Prospective Register of Systematic Reviews (PROSPERO) (CRD42020208449).

Criteria for considering studies in this review

We included randomized clinical trials (RCTs) and observational studies (cross-sectional, longitudinal, case-control, and cohort) of adult patients with a confirmed diagnosis of COVID-19 published between 01/12/2019 and 10/09/2020. All editorials, reviews, clinical cases, and in vitro studies were excluded. The studies included were aimed at assessing ADL after COVID-19 infection.

Search strategy

We reviewed the Embase, Cochrane Library, CINAHL, Web of Science, and PubMed/MEDLINE databases on September 10, 2020 using the following search terms. For condition: SARS-CoV-2 OR COVID-19 OR 2019 novel coronavirus infection OR COVID19 OR coronavirus disease 2019 OR coronavirus disease-19 OR 2019-nCoV disease OR 2019 novel coronavirus disease OR 2019-nCoV infection. For the main outcome: Functional capacity OR activities of daily living OR Functional independence measure OR Barthel index OR Lawton Brody scale OR Katz index OR ADL OR PCFS OR Functional scale OR Functional status (Supplemental File 1).

All references were exported to Rayyan Web software (Ouzzani et al., 2016); we then conducted a bibliographic search and compiled the identified records and removed any duplicated studies. Two authors (CSR and CPP) independently reviewed the information to extract data from the included studies. All included studies were assessed for internal validity using a checklist.

Reviewing procedure and study selection

The reviewing procedure was performed in two steps: (1) The titles and abstracts of all references were reviewed by two investigators (CSR and CPP). Studies deemed not relevant based on the review of the title and abstract were excluded. Conflicts were solved by a third reviewer (RTC). (2) The articles selected in the first step were read in full-text version and checked again using the eligibility criteria (CSR and CPP). Any disagreements were solved by a third reviewer (RVU).

Data extraction

Two authors (CSR and CPP) extracted the data independently and in duplicate using a standardized protocol and reporting forms. For each study, instruments, characteristics of the patients, main outcome results, time of assessment, success in evaluation and conclusions were extracted. If any relevant data were not in the article, the author was contacted to request the information.

Risk of assessment

The risk of bias and the quality of observational and interventional studies was assessed using the corresponding assessment tools recommended by the National Heart, Lung & Blood Institute (2020). Each tool contains criteria against which internal validity and risk of bias are evaluated. The criteria were evaluated as “Yes”, “No”, or “Other” (not reported, not applicable, or not determinable), and an overall rating was provided for each study based on the items rated with an affirmative answer: ≥75% = good, 50–75% = fair, <50% = poor. Two investigators (CSR and CPP) were used to perform this assessment, while disagreements were solved by arbitration by a third investigator (LVC).

Results

The initial search found a total of 1,228 studies, 1,005 studies remained after removing duplicates. After abstract and title screening, 983 studies were excluded. Twenty-one full texts remained, which were assessed for eligibility leading to the exclusion of 12 studies due to wrong publication type (n = 5), research protocol (n = 3), wrong study design (n = 2), incomplete data (n = 1), and preprint (n = 1). The process was embodied in a flowchart (Fig. 1). Nine studies (Halpin et al., 2020; Belli et al., 2020; Curci et al., 2020; Sakai et al., 2020; Liu et al., 2020; Bousquet et al., 2020; Zerah et al., 2020; Mcloughlin et al., 2020; Agarwal et al., 2020) met all the inclusion criteria. Four retrospective (Belli et al., 2020; Sakai et al., 2020; Zerah et al., 2020; Agarwal et al., 2020), 2 prospective (Bousquet et al., 2020; Mcloughlin et al., 2020), 2 cross-sectional (Halpin et al., 2020; Curci et al., 2020) and one RCT (Liu et al., 2020) studies were included. A total of 1,465 patients were studied in the included articles. The median age was 68.98 years (± 8.29) of which 716 (48.9%) were women. In total 1,190 (81.2%) were inpatients (Table 1).

Figure 1 Flowchart of included studies.

Table 1 Characteristics of included studies.

Author, year
country	Design	Participants (F/M)	Age (years)	BMI/Anthropometrics	Setting/Intervention	Patients characteristics	
Liu et al. (2020)
China	RCT	CG: 36 (12/24)
IG: 36 (11/25)	CG: 68.9 ± 7.6
IG: 69.4 ± 8.0	CG: 22.9 ± 3.9
IG: 23.1± 3.5	Outpatients/RRP	DLCO% predicted CG: 60.7 ± 12.0
DLCO% predicted IG: 60.3 ± 11.3
FEV1/FVC (%): CG: 60.44 ± 5.77; IG: 60.48 ± 6.39
CT multilobar lesion 48 (66.6%)
Comorbidities: HT: 18 (25%); DM: 18 (25%)	
Bousquet et al. (2020) France	Prospective	108 (48/60)	78 ± 7.8	17 (16%)
<21 kg/m2	Inpatients/NI	CT scan lung damage: 82 (76%) mild 21 (19.4%), moderate 35 (32.4%), extensive 20 (28.5%), severe 6 (5.5%). Mortality rate: 28 (26%).
Comorbidities: HT (71%), DM (28%)	
Sakai et al. (2020)
Japan	Retrospective	43 (12/31)	65 (range 21–95)	NR	Inpatients and outpatients/RRG vs. DRG	Required ventilation: 12 (27.9%), ECMO: 1 (2.3%), required oxygen: 22 (51%), asymptomatic or mild: 9 (20.9%)	
Zerah et al. (2020) France	Retrospective	821 (476/345)	86 ± 7	NR	Inpatients/NI	Severe at admission: 107 (13%), died: 250 (31%), discharged: 159 (19%), transferred to rehabilitation centers: 365 (45%), ≥2 comorbidities:
698 (85%)	
Curci et al. (2020)
Italy	Cross-sectional	32 (10/22).	72.6 ± 10.9	24.7 ± 2.4	Rehabilitation unit/rehabilitation	FIO2 ≥21% and <40%: 13 (40.6%)
FIO2 ≥40% and <60%: 19 (59.4%)	
Belli et al. (2020)
Italy	Retrospective	103 (50/53)	73.9 ± 12.9	26.6 ± 5.8	Inpatients/PR	No MV: 82 (79.6%),
NIV: 9 (8.7%),
MV and NIV: 12 (11.7%).
O2-supplementation: 90 (78.6%), ≥2 comorbidities: 91 (88%), and ≥4: 57 (55%)	
Mcloughlin et al. (2020)
UK	Prospective	71 (18/53)	61 (range 24–91)	NR	Inpatients/NI	Acute medical wards 25 (35.2%),
High Dependency Unit 5 (7%)
Critical care 41 (57.7%)	
Agarwal et al. (2020) USA	Retrospective	115 (33/82)	63.5 ± 8.6	LCM: 28.7 ± 5.9
OMRIBF: 27.4 ± 6.1	Inpatients/NI	LCM patients 35 (30.4%) required longer ventilator support and were more likely to have a moderate and severe ARDS score	
Halpin et al. (2020)
UK	Cross-sectional	100 (46/54)	64.5 (range 20–93)	OW-OB: 59
HW: 25 UW: 3
Unknown: 11	Outpatients (discharge)/NI	ICUP group: 32 (32%)
WP group: 68 (68%)	
Notes:

Data are shown as Mean ± SD, Median (Inter-quartile range), n (%)

Abbreviations: ARDS: Acute Respiratory Distress Syndrome; CG: control group; CT: Computed tomography; DLCO: Diffusing Capacity of the Lungs for Carbon Monoxide; DM: diabetes mellitus; DRG: direct rehabilitation group; ECMO: Extracorporeal Membrane Oxygenation; FEV1: Forced expiratory volume at 1 s; FIO2: Fraction of inspired oxygen; FVC: Forced vital capacity; HT: hypertension; HW: Healthy weighted; IG: intervention group; ICUP: intensive care unit patients; LCM: Leukoencephalopathy and/or Cerebral Microbleeds; MV: Mechanical Ventilation; NI: no intervention; NR: not reported; NIV: Non Invasive Mechanical Ventilation; OMRIBF: Other Magnetic Resonance Imaging Brain Findings; OB: obese; OW: Overweighted; PR: Pulmonary Rehabilitation; RCT: randomized controlled trial; RRG: remote rehabilitation group; RRP: Respiratory rehabilitation program; UW: Underweighted; WP: ward patients.

The included studies used different instruments such as BI (assesses BADL) (Belli et al., 2020; Curci et al., 2020; Sakai et al., 2020), ADL score (assesses BADL and IADL) (Bousquet et al., 2020; Zerah et al., 2020), FIM (assesses BADL in motor and cognitive domain) (Liu et al., 2020), Composite Functional Score (CFS) (assesses BADL and IADL) (Mcloughlin et al., 2020), mRS (assesses disability in BADL and IADL) (Agarwal et al., 2020) and EQ-5D-5L (assesses BADL) (Halpin et al., 2020). For each study, design, sample size, gender, age, body mass index, setting or intervention and main patient characteristics were extracted (Table 1).

A quality assessment was performed using the National Heart, Lung and Blood Institute’s study assessment tools. Six (66.6%) of the nine studies included obtained a “Good” quality ranking with a total score of over 75% (Curci et al., 2020; Liu et al., 2020; Bousquet et al., 2020; Zerah et al., 2020; Mcloughlin et al., 2020; Agarwal et al., 2020) while three (33.3%) of them pointed to a “Fair” quality ranking with a total score of between 50% and 75% (Halpin et al., 2020; Belli et al., 2020; Sakai et al., 2020). No “Poor” quality ranked studies were included (Supplemental File 2).

Main results

Barthel index

BI was assessed in three studies (Belli et al., 2020; Curci et al., 2020; Sakai et al., 2020). The main objective was to assess functional ability (Belli et al., 2020) and assessment of early rehabilitation programs for patients with COVID-19 (Curci et al., 2020; Sakai et al., 2020). Measures were carried out on various occasions, such as on admission (Curci et al., 2020), discharge (Sakai et al., 2020) or both (Belli et al., 2020). The results of the studies showed low BI scores in this group of patients, which is related to a worse level of functional ability. Belli et al. (2020) reported 67% of patients with <60 points (poor score) on the BI, and 45.6% of patients who were bedridden when admitted to a rehabilitation hospital. At discharge, the scores were significantly better with 47.5% of the patients still scoring poor on the BI, and only 17.5% who were still bedridden (Belli et al., 2020). Curci et al. (2020) exhibited differences in the scores of BI on higher and lower Fraction of Inspired Oxygen (FiO2) patient groups, showing lower scores on the higher FiO2 group at ICU discharge. Moreover, Sakai et al. (2020) indicated differences in mean scores regarding different rehabilitation groups (in-person vs. remote rehabilitation). The in-person rehabilitation group was significantly older, had more intubated individuals, and had worse mean scores at baseline and discharge, 40 vs. 70, respectively.

In contrast, the remote rehabilitation group had a higher mean BI at baseline and had no change with the intervention, 90 vs. 90, respectively. None of the 3 studies reported the results of each item of the BI, except for Sakai et al. (2020), who reported mobility score. Together, these results show that BI allows us to identify that a worse individual context is related to low ADL functional ability previous to COVID-19 and is also related to a worse score post COVID-19. At the same time, BI is able to detect the worsening in ADL in around 60% of subjects after COVID-19.

ADL score

The ADL score tool was used in two studies to determine functional ability status in older patients and the relationship with prognostic factors and mortality associated with COVID-19 (Bousquet et al., 2020; Zerah et al., 2020). These studies focused on the ADL functional ability status at the beginning of hospitalization. Six basic tasks related to BADL were assessed: bathing, dressing, toileting, transfer, continence, and feeding. Zerah et al. (2020) reported a significant difference in older patients with COVID-19 between survivors and non-survivors, with a median of 4.5 (IQR 2-6) and 3 (IQR 1-6) BADL respectively. Bousquet et al. (2020) showed that 50% of patients had altered at least 1 BADL, and also evaluated four IADL (using the telephone, transport, medications, and money management) which showed that 63% of patients had altered at least 1 IADL. Results indicated that a decreased functional status and dependency in older patients were associated with short-term mortality. In both studies, the score for each ADL was not reported. These results show that the ADL score can identify the impact of ADL functional ability status in older patients and the negative outcomes related to high patient vulnerability.

Functional independence measure

FIM was assessed in one study to investigate the effects of respiratory rehabilitation on respiratory function, ADL, QoL, and psychological status in elderly patients with COVID-19 who had been discharged from hospital (Liu et al., 2020). The evaluation was carried out at discharge and at 6 weeks in the post rehabilitation program. Older subjects with moderate or severe dementia, previous neurological, respiratory or cardiovascular disease were excluded. Baseline Mean FMI scores (pre-intervention) were around 109 (out of 126) for both groups. There was no significant improvement post-intervention in the intervention group nor compared with the control group. Results of each FIM item were not reported. In contrast to previous works, assessment of ADL functional ability with FIM showed patients that require minimal assistance; the authors did not find any improvement with the intervention.

Composite functional score

CFS was performed on one study to determine functional and cognitive outcomes among patients with delirium in COVID-19 using a composite of the BI plus Nottingham Extended Activities of Daily Living (NEADL) (Mcloughlin et al., 2020). The evaluation was carried out during hospital stay and 4 weeks after delirium was ascertained. Results of CFS showed worse physical function in post delirium patients (97 vs. 153, p < 0.01), but the results of each BI and NEADL item were not reported. In this way, for COVID-19 hospitalized patients without delirium the CFS showed high values of independence after COVID-19. In contrast, in patients with delirium CFS was able to detect poor functional ability outcomes (Mcloughlin et al., 2020).

Modified Rankin scale

mRS was used in one study to correlate cerebral microbleeds and leukoencephalopathy with clinical, laboratory, and functional outcomes in adult and elderly inpatients with COVID-19 (Agarwal et al., 2020). All patients reported worse functional ability in the mRS assessment, but patients with leukoencephalopathy and/or cerebral microbleeds required longer ventilator support, had longer hospitalization stays, and higher mRS scores at discharge compared with patients without these magnetic resonance imaging findings. The higher mRS scores indicate overall worse functional status on discharge. In this way, mRS assessment was able to detect worsening ADL functional ability post COVID-19. Moreover, mRS detected differences in functional ability related to other clinical variables related with the severity of the disease.

EQ-5D-5L

EQ-5D-5L was used in one of the included studies to assess the impact of pre- and post- COVID-19-disease on each participant’s mobility, personal care, usual activities, pain and anxiety/depression 4–8 weeks after hospital discharge (Halpin et al., 2020). The results showed a significant drop in EQ-5D-5L score in 68.8% of ICU patients and 45.6% in patients from other wards. Patients presented a clinically significant decrease in EQ-5D-5L on average 48 ± 10.3 days post discharge, which translated into worsened mobility, self-care, usual activities, pain/discomfort, and anxiety/depression. These results show that the EQ-5D-5L is able to identify the impact of COVID-19 on ADL functional ability status after discharge. Also, EQ-5D-5L is able to detect differences related to the severity of the disease.

Discussion

This review identified nine studies that assessed performance of ADL on COVID-19 patients using eight different scales: BI (Belli et al., 2020; Curci et al., 2020; Sakai et al., 2020), ADL score (Bousquet et al., 2020; Zerah et al., 2020), FIM (Liu et al., 2020), Composite Functional Score (CFS) (Mcloughlin et al., 2020), mRS (Agarwal et al., 2020) and EQ-5D-5L. In these studies, the performance of ADL on COVID-19 patients was mainly used to explore the association between baseline functional ability as a mortality predictor (Bousquet et al., 2020; Zerah et al., 2020) and to identify changes in functional ability outcomes after the acute period of infection (Mcloughlin et al., 2020; Agarwal et al., 2020) (Table 2).

Table 2 Findings of included studies.

Author, Year	Instrument	Results	Time assessment	Success in the evaluation, n (%)	Conclusions	
Liu et al. (2020)	FIM	Pre-Intervention:
FIM score CG: 109.3 ± 10.7; IG: 109.2 ± 13
Post-intervention:
FIM score CG: 108.9 ± 10.1; IG: 109.4 ± 11.1	Before and after pulmonary rehabilitation	72 (78.2%)	Six-week respiratory rehabilitation can improve respiratory function, QoL, and anxiety, but does not improve the FIM score of elderly patients with COVID-19	
Bousquet et al. (2020)	ADL Score	ADL score ≤ 5/6: 54 (50%)
IADL score ≤ 3/4: 68 (63%)	Before hospitalization	108 (90%)	ADL-dependency before hospitalization, serum levels of D-Dimers and LDH, and the absence of anticoagulation were the factors independently associated with 1-month mortality in older inpatients with COVID-19	
Sakai et al. (2020)	BI	RRG: BI mobility score 15; total score 90
DRG: BI mobility score 10 (range 0–15); total BI score 70 (range 0–85)	At hospital discharge (before and after rehabilitation)	43 (97.7%)	Both mobility and total BI scores improved in both groups after intervention	
Zerah et al. (2020)	ADL Score	Mean ADL score 4 (IQR2-6)
ADL score (Non-survivors) 3 (IQR 1–6)
ADL score (Survivors) 4.5 (IQR 2–6)	During hospital stay	821 (93.5%)	Hospital mortality was associated with lower ADL scores (ADL < 4)	
Curci et al. (2020)	BI	Total BI score: 45.2 ± 27.6
FIO2 ≥ 21% and <40% (n = 13)
BI score 53.3 ± 29.3
FiO2 ≥ 40% and <60% (n = 19)
BI score: 39.6 ± 25.7	At admission to the rehabilitation unit	32 (88.8%)	COVID-19 patients had severe disability. Only 14 were able to walk	
Belli et al. (2020)	BI	At entry rehabilitation institute: 67%: ≤60 BI score
At discharge rehabilitation institute: 47.5%: ≤60 BI score	At entry and discharge of rehabilitation institute	103 (89.5%)	Physical functioning and performance of ADLs were still significantly impaired at discharge to their home	
Mcloughlin et al. (2020)	CFS	Pre Delirium: 153/166 CFS score
Post Delirium: 97/166 CFS score	During hospital stay and 4 weeks after hospital discharge	71 (86.59%)	Delirium was associated with functional impairments in the mid-term	
Agarwal et al. (2020)	mRS	LCM: 5 (IQR 4–5) mRS score
OMRIBF: 4 (IQR 2–5) mRS score	At hospital discharge	115 (100%)	LCM was associated with critical illness, mortality and worse functional outcome in patients with COVID-19	
Halpin et al. (2020)	EQ-5D-5L	22ICUG participants (22%) experiencing new problems in mobility, self-care or usual activities.	Mean 48 ± 10.3 days post hospital discharge	100 (100%)	COVID-19 was associated with illness-related fatigue, breathlessness and psychological distress leading to a significant drop in quality of life	
Note:

Abbreviations: ADL: Activities of daily living; BI: Barthel Index; CFS: Composite Functional Score; CG: control group; CT: computerized tomography; DRG: direct rehabilitation group; FIM: Functional Independence Measure; IADL: Instrumental Activities of daily living; ICUG: Intensive care unit group; IG: intervention group; IQR: Inter-quartile range; LCM: Leukoencephalopathy and/or Cerebral Microbleeds; LDH: Lactate Dehydrogenase; mRS: Modified Rankin Scale; NI: no intervention; NR: not reported; OMRIBF: Other Magnetic Resonance Imaging Brain Findings; qSOFA: quick Sequential Organ Failure Assessment score; RRG: remote rehabilitation group.

In this way, ADL assessment can be used to identify functional limitations, to understand the patient’s prognosis and to evaluate the intervention’s effect on the subject. The majority of studies included in this review evaluated ADL: (1) On admission to a rehabilitation unit, mostly as a prognostic factor, and (2) At discharge and during the follow-up period, to evaluate the impact of the disease but also to compare the progress obtained as result of an intervention or rehabilitation process.

In all of the studies included, the findings revealed a decline in ADL performance after COVID-19 infection regardless of the scale applied. Those who had the worse results were older patients and/or patients who had complications during their hospital stay, such as being admitted to ICU (Halpin et al., 2020), mechanical ventilation (Belli et al., 2020), delirium (Mcloughlin et al., 2020), cerebral microbleeds, leukoencephalopathy (Agarwal et al., 2020) or greater oxygen requirements (Curci et al., 2020). Participation in a rehabilitation program was also a factor likely to modify the outcomes in functional capacity, however, not all cases were able to improve the performance of ADL (Liu et al., 2020). Several factors regarding a patient’s context are likely to affect the success, or lack of it, in improving the functional capacity and ADL performance of the subjects.

Our results show the different tools used to evaluate functional abilities during ADL. However, not all evaluate the same dimensions. Among the tools used are the BI and the FIM (Belli et al., 2020; Curci et al., 2020; Liu et al., 2020; Sakai et al., 2020). The BI assesses the performance in BADL, such as feeding, personal toileting, bathing, or dressing, with a score between 0 and 100 points (Bouwstra et al., 2019). A higher number reflects greater ability to function independently and has the advantage that it can be applied to self-reporting or direct administration (Bouwstra et al., 2019). The FIM assesses the functional status with scores which range from 18 (lowest) to 126 (highest), but there is a difference to BI, which in addition to BADL incorporates psychological and social functions as communication or special cognition (Maritz et al., 2019). Both instruments are widely used, particularly in patients with neurological conditions (Bouwstra et al., 2019; Maritz et al., 2019).

The EQ-5D-5L is a variation of the EQ-5D, a standardized instrument for measuring generic health status (Janssen et al., 2013). This instrument assesses the mobility, self-care, usual activities, pain/discomfort and anxiety/depression, allowing the evaluation of BADL (Janssen et al., 2013). The ADL score assesses washing, getting dressed, moving about indoors, going to the toilet, eating, and continence with scores ranging from 0 to 6. A total score ≤ 5/6 indicates ADL-dependency (Burton & Potter, 2017). It is likely that the simplest test is the mRS, which measures the degree of disability or dependance in the ADL of people who have suffered a stroke or other causes of neurological disability (Broderick, Adeoye & Elm, 2017). However, it does not allow precise discrimination of the dimensions affected when planning an intervention. On the other hand, if a tool is not capable of meeting the clinician’s requirements, composite tools can be used, as was the case in one of the selected studies.

ADL performance acts as an indicator of a patient’s prognosis (Bousquet et al., 2020; Zerah et al., 2020; Level et al., 2018). Complementing rehabilitation efforts on ADL training may lead to a major impact on the patient’s life, rather than just considering physical function parameters (Elliott et al., 2011). The importance of ADL as a prognostic factor can be seen in the present association between dependance or low functional capacity with short- and medium-term mortality (Guidet et al., 2020).

Another aspect related to independence level is cognitive performance, which can be included to complement the assessment of ADL (Torres-Castro et al., 2021). The cognitive sphere is particularly relevant in IADL due to higher complexity invested on the involved tasks at all age groups and particularly on elderly people. Therefore, tools such as the Mini-Mental State Examination (MMSE) or Montreal Cognitive Assessment (MOCA) could complement ADL evaluation. These tests are highly cost-effective since they are easy to implement by healthcare professionals in all clinical settings. Other relevant variables to keep in mind for a more complete patient assessment when COVID-19 survivors are comprehensively evaluated is respiratory function, quality of life and physical capacity (Torres-Castro et al., 2021). In particular, we must have a special focus on respiratory function, due to COVID-19 being mainly a respiratory disease and which has reported sequelae in lung function after infection (Torres-Castro et al., 2020). We consider that these should be evaluated more exhaustively in further long-term studies.

Patients are out of their normal context during hospitalization, thus some ADL evaluations, especially IADL, must be adapted or omitted according to the nature of the assessment. Therefore, ADL evaluations should be completed once the patient is discharged and returns to his context so that the impact of COVID-19 after the acute phase can be more accurately estimated. In addition to the context, it becomes cardinal to select the adequate assessment tool to obtain accurate results (Bai et al., 2020).

In the literature we find a specific tool to assess functional status, the Post-COVID-19 Functional Status (PCFS) Scale, which could be used for patient follow-up (Klok et al., 2020). This scale assesses the patient’s functional capacity upon discharge from the hospital and allows functional sequelae to be assessed. Although it does not strictly correspond to a test that evaluates ADL, its information can be useful to complement the functional evaluation of these patients, especially since studies have already appeared that have validated it in post-COVID-19 patients (Lorca et al., 2020).

In this work, we identified six tests used to explore the independence level during ADL to estimate the impact in functional ability and the relationship with prognostic factors and mortality associated with COVID-19. A key aspect of integrating information from different studies requires knowing the features of the sample related to intrinsic capacity and context described in each study. Thus, in the studies, including subjects with severe intrinsic capacity impairment, most of the tests are suitable for exploring the independence level during ADL, independently of the context. A different scenario could be observed in studies conducted in a rehabilitation program context (leading to selection bias). Liu et al. (2020) reported a higher level of independent functional ability post COVID-19 in comparison with samples coming from studies conducted in other different contexts not related to rehabilitation. In this way, the combination of intrinsic capacity and context gives us valuable information which can be used to select the appropriate assessment tool.

Until specific instruments that assess ADL appear and are validated in post-COVID-19 patients, we recommend the use of tools designed for this purpose, such as the BI, the FIM or the ADL Score. Although the FIM did not show any discrimination in the effect of an intervention (Liu et al., 2020), we recommend that future studies use the FIM in a population with greater functional compromise. Further studies, including a deeper analysis of ADLs, both BADL and IADL, are required to explore the impact of COVID-19 on functional performance among adults, especially on elderly people.

Limitations and strengths

The limitations of this review were mostly related to the heterogeneity of the studies, including patient characteristics, study design, methods, and assessment timeframe; in most of the studies there is no detailed description of demographic and clinical features of the sample. Also, we reviewed a small number of studies which only show short-term outcomes (long-term are unknown), and ADL/functional performance were not the main aim of the studies. Another important limitation is that our results are based on articles where the mean age of people was over 60 years old, so the findings cannot be extrapolated to people under that age. The main strength is its focus on the assessment of ADL as an essential outcome to optimize future rehabilitation protocols and resource allocation. Furthermore, it provides an overview of various assessment tools that have been used to explore ADL among COVID-19 patients.

Conclusions

All included studies found a reduction of ADL despite the test or scale used, revealing a vital worsening of physical function, deterioration of ADL performance and consequently loss of independence of COVID-19 patients after the acute phase of infection. Besides, the functional ability status previous to COVID-19 is vital for predicting the severity of the disease and mortality. Considering the results shown in our work, the choice of the best test to assess ADL is closely related to the intrinsic capacity and the subjects’ context. Thus, Barthel Index and ADL score were the most used assessments across subjects with different levels of intrinsic capacity and contexts.

Assessment of activities of daily living provide relevant information regarding the functional impact on COVID-19 patients, which contributes towards identifying the rehabilitation needs of this population and could inform the allocation of resources to support their recovery.

Well-designed studies should research short and long-term functional impact on post-COVID-19 patients with the objective of optimizing intervention strategies and supporting decision-making in clinical practice. It is also essential to further test the psychometric properties of these tests on COVID-19 patients. The data collected to date in this systematic review is a useful starting point for further studies.

Supplemental Information

Supplemental Information 1 PRISMA checklist.

Click here for additional data file.

Supplemental Information 2 Search Strategy.

Click here for additional data file.

Supplemental Information 3 Quality Assessment.

Click here for additional data file.

Supplemental Information 4 Rationale and contribution.

Click here for additional data file.

Additional Information and Declarations

Competing Interests

Author Contributions

Data Availability

The authors declare that they have no competing interests.

Catalina Pizarro-Pennarolli conceived and designed the experiments, performed the experiments, analyzed the data, prepared figures and/or tables, authored or reviewed drafts of the paper, and approved the final draft.

Carlos Sánchez-Rojas conceived and designed the experiments, performed the experiments, analyzed the data, prepared figures and/or tables, authored or reviewed drafts of the paper, and approved the final draft.

Rodrigo Torres-Castro conceived and designed the experiments, performed the experiments, prepared figures and/or tables, authored or reviewed drafts of the paper, and approved the final draft.

Roberto Vera-Uribe conceived and designed the experiments, performed the experiments, authored or reviewed drafts of the paper, and approved the final draft.

Diana C. Sanchez-Ramirez conceived and designed the experiments, performed the experiments, prepared figures and/or tables, authored or reviewed drafts of the paper, and approved the final draft.

Luis Vasconcello-Castillo analyzed the data, prepared figures and/or tables, authored or reviewed drafts of the paper, and approved the final draft.

Lilian Solís-Navarro analyzed the data, authored or reviewed drafts of the paper, and approved the final draft.

Gonzalo Rivera-Lillo analyzed the data, authored or reviewed drafts of the paper, and approved the final draft.

The following information was supplied regarding data availability:

This is a systematic review and all data are in Tables 1 and 2.

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
