# Peer review of "Assessment of activities of daily living in patients post COVID-19: a systematic review"

_PeerJ, doi:10.7717/peerj.11026_

## Round 0.1 · original submission · Minor Revisions

Thank you for your timely paper. Please refer to the reviewers' comments when making your revisions.

Reviewer 1 ·

Basic reporting

The article is well written.
2-But These suggestions may help readers to better understand the article.
3-The conclusion in the abstract is vague. It is not your article's conclusion.
4- Introduction is well written, but it is too long.

Experimental design

no comment

Validity of the findings

no comment

Additional comments

5- some of your references are old Like (Weber-Carstens, 2011).
6- in covid-19 do we have muscle weakness or we have myalgia. Please explain what will happen in covid-19 in line 19-21.
7- please in conclusion explain about the differences between ADL in hospitalized patients and Before hospitalization and at discharge patients.

Reviewer 2 ·

Basic reporting

1. In table2, the time of measurements should be more specific. Please specify if the at admission/discharge means admission/discharge of hospital or rehabilitation facility.
2. In page 13 line 1-2, the reference should be inserted regarding on the statement of ‘In COVID-19 patients, delirium was found to be prevalent and was associated with poor functional outcomes.’

Experimental design

1. Most of the studies included in this paper were using different instruments to measure the ADLs, thus it is suggested that the authors should add some descriptions about these measurements, for example, what is the focus of each instruments and the differences among these instruments? What is the criteria of being ‘good’ or ‘bad’?
2. In page 11 line 15, the authors stated that one of the studies ‘reported 67% of patients with <60 points and 45.6% of patients who were bedridden’. However, the measurements should be more clearly stated. The authors stated that the measures in this study were carried out at both admission and discharge; can the authors provide the information about when these scores were measured? Are these percent changes or raw percentages?
3. In page 11 line 18, the authors wrote ‘the direct rehabilitation group was significantly older, had more intubated individuals, and had worse scores at baseline and discharge’, however, the reference group should be specified. Otherwise it is not clear why the authors would like to report the data of direct rehabilitation group and how does this relate to the impact of COVID-19 on ADL performance. In addition, the authors mentioned that the Sakai et al. study measured the BI scores at discharge, however in line 20, the authors reported the results both at baseline and discharge, this is a little confusing.

Validity of the findings

1. In page 15 line 3, since the conclusions are not consistent between the studies, can the authors comment on the possible reasons/factors that could possibly affect the results?
2. The severity of COVID might have impacts on the ADL performance, can the authors provide more information on the severity of disease for each of the studies included?
3. The heterogeneity of these included studies is large. The studies were using different measurements and each of the instruments were measured at different times based on different populations, which might weaken the conclusions. Thus, the authors should discuss more about the possible impacts on these differences, and why they could be combined together.

Reviewer 3 ·

Basic reporting

COVID-19 has caused global health and economic crisis. This paper investigated the impact of the disease on activities of daily living (ADL) performance of COVID-19 infection by a systematic review, which will interest a wide variety of readers. However, from the introduction, it is unclear the gap of current knowledge and how a systematic review can fill the gap. In some parts, the statement is ambiguous. For example lines 97-102, to my understanding, “the great challenge” is “a variable number of subjects” can finish the tests, but how a “further analysis” can overcome this challenge?

Experimental design

In the results part, the authors searched, evaluated, and selected the available studies. Line 217, “a total of 1465 patients were studied in the articles included”. However, if the authors could not integrate information from different studies, they failed to reach high statistical power and robust results by large numbers of patients.
Line 243-244, do the bedridden patients have a worse score?
Line 246 to 248, different rehabilitation groups have different scores. Is that because more severe patients need rehabilitation? Can rehabilitation improve a patient’s score?
Line 254-259, the two studies accessed the ADL scores at a different timeline. Do they have different findings?
Line 264-265, “the evaluation was carried out at discharge and six weeks later,” Did the score improve six weeks later compared to at discharge?
Line 321-323, what is the underlying reason that rehabilitation can not improve ADL?

Validity of the findings

The conclusion for most result part is missing or unclear.
COVID19 is mainly a respiratory disease. How impaired respiratory function affect the ADL of the patients?

Reviewer 4 ·

Basic reporting

The article deals with a very topical issue. The literature produced so far has partially investigated the impact of the disease on the activities of daily life.
About 1500 patients were investigated, with articles from all over the world; during the selected research period the articles to be considered were those highlighted by the authors.
Tablel 1: setting is “inpatient” in Belli et al's article. Not outpatients. Rehabilitation Clinic transformed in acute setting without ICU for positive patients.

Experimental design

The proposed review is in accordance with the journal's aim and scope.
Research question is defined
The survey was conducted correctly according to correct methods of investigation.
The methodology of the research was carried out according to a correct qualitative method (National Heart, Lung and Blood Institute’s study assessment tools).
.

Validity of the findings

The following work can help the reader to quickly overview what has been achieved so far.
I would advise authors to introduce a part of suggestions for further investigation at the end of the discussion. What is the best evaluation tool to use? What should be the rationale for further evaluating ADLs? What could be the future research fields? How can we do a deeper analysis of ADLs?
As this is a systematic review, is it necessary to provide additional data to support that covid has a negative impact on daily life?
This work underline an heterogeneity of the methodologies used for the evaluation of ADLs. which is the best one?
We still know little about the impact of the disease on younger patients still of working age. Can we use the same tools?

Additional comments

Interesting argument that further underlines the impact of covid-19 during daily life activities.
1-On the basis of the articles highlighted, do the authors consider the literature so far produced to be exhaustive?
2-"The data collected to date in this systematic review is a useful starting point for further studies" :the authors could suggest the best strategies to follow for future research.
3-the strengths are contained in the chapter "limtations"; it may be useful to split them or rename the chapter "Limitations and Strengths"

---

## Round 0.2 · accepted · Accept

I am pleased to inform you that your manuscript has been accepted.

Reviewer 2 ·

Basic reporting

no comment

Experimental design

no comment

Validity of the findings

no comment

Additional comments

Thanks for addressing my questions. The article looks great.

Reviewer 3 ·

Basic reporting

The comments were properly addressed.

Experimental design

The comments were properly addressed.

Validity of the findings

The comments were properly addressed.

Additional comments

Thank you for properly addressing the comments!